# Video-Rate Quantitative Phase Imaging Using a Digital Holographic Microscope and a Generative Adversarial Network

**DOI:** 10.3390/s21238021

**Published:** 2021-12-01

**Authors:** Raul Castaneda, Carlos Trujillo, Ana Doblas

**Affiliations:** 1Department of Electrical and Computer Engineering, The University of Memphis, Memphis, TN 38152, USA; rcstdqnt@memphis.edu; 2Applied Optics Group, Physical Sciences Department, Universidad EAFIT, Medellin 050037, Colombia; catrujilla@eafit.edu.co

**Keywords:** phase compensation, digital holographic microscopy, learning-based method, generative adversarial networks, video-rate performance

## Abstract

The conventional reconstruction method of off-axis digital holographic microscopy (DHM) relies on computational processing that involves spatial filtering of the sample spectrum and tilt compensation between the interfering waves to accurately reconstruct the phase of a biological sample. Additional computational procedures such as numerical focusing may be needed to reconstruct free-of-distortion quantitative phase images based on the optical configuration of the DHM system. Regardless of the implementation, any DHM computational processing leads to long processing times, hampering the use of DHM for video-rate renderings of dynamic biological processes. In this study, we report on a conditional generative adversarial network (cGAN) for robust and fast quantitative phase imaging in DHM. The reconstructed phase images provided by the GAN model present stable background levels, enhancing the visualization of the specimens for different experimental conditions in which the conventional approach often fails. The proposed learning-based method was trained and validated using human red blood cells recorded on an off-axis Mach–Zehnder DHM system. After proper training, the proposed GAN yields a computationally efficient method, reconstructing DHM images seven times faster than conventional computational approaches.

## 1. Introduction

Quantitative phase imaging (QPI) techniques quantify the phase measurements from micrometer-sized biological and non-biological samples. Because phase measurements encode sample’s topography and refractive index information, QPI methodologies have been widely utilized for diagnosing diseases and refining manufacturing processes [1,2,3,4]. Among the reported QPI techniques, methods based on the transport equation [5,6], the Hilbert transform [7,8,9], the diffraction of white light [10,11], and digital holographic microscopy (DHM) [12,13,14] are counted among the most utilized. The DHM systems are based on optical interferometry to reconstruct both amplitude and phase distributions of biological and non-biological specimens, providing functional and morphological sample information. Owing to its high sensitivity, imaged field of view, and frame rate acquisition, DHM stands out among the QPI techniques to reconstruct high-resolution phase images from unstained samples [13]. The robustness and sensitivity of the phase measurements in some DHM systems within the nanometric range have allowed dynamic imaging, for instance, three-dimensional (3D) particle tracking [15,16], cell motility studies [17,18,19], and dynamic changes of surface topography [20,21].

Over the last decade, DHM has become a mature technology thanks to thousands of research studies that have focused on investigating its optical design, phase reconstruction algorithms, and potential applications of this modality to life and materials sciences [22,23,24,25]. Despite the successful performance of DHM systems, its applicability to in situ clinical research has been partially hampered by the need for a standard phase reconstruction algorithm that provides quantitative phase distributions without any phase distortion [13]. The presence of phase aberrations may provide inaccurate results, leading to erroneous conclusions about specimen properties. Accurate phase measurements are imperative since variations in phase measurements are used as a diagnostic and measuring tool in life sciences [12,13,14]. The DHM technology retrieves this phase distribution after applying a computational reconstruction approach. At present, individual DHM research groups have developed and implemented their numerical algorithms to reconstruct their phase images. Because a standard reconstruction method does not exist, different phase values may be measured using different reconstruction methods and implementations for the same sample and DHM system. The off-axis image-plane DHM system operating in the telecentric regime is considered to be the simplest DHM system, since it requires the minimum number of numerical processes [25]. Traditionally, phase measurements are reconstructed using the Fourier filtering approach. However, the implementation must be precisely executed to avoid imprecise phase measurements. The numerical process in off-axis image-plane DHM systems operating in telecentric regime involves (i) the proper selection of the frequency components carrying the information of the sample in the Fourier domain of the acquired DHM image and (ii) the phase compensation of the tilting angle between the interfering waves due to the off-axis configuration. Incorrect selection of frequency components may lead to low-resolution phase measurements (i.e., phase images that are not limited by diffraction), and inaccurate tilting angle compensation can introduce errors in the quantitative phase measurements. One straightforward compensation method for precise phase extraction is manually selecting the tilt until obtaining a phase reconstruction without sawtooth fringes. Nonetheless, choosing the best reconstructed phase image can be challenging for the end-user in DHM, since it depends significantly on the user’s expertise. The DHM reconstruction algorithm should be automatic and adaptable to any sample and imaging conditions to increase its applicability in life and materials studies. Whereas several automated DHM reconstruction approaches [26,27,28,29] have been proposed; often these proposals yield reconstructed phase images with phase nuisances, preventing any quantitative analysis. In addition, the computational complexity of all these methods still restricts the proper recovery of phase maps free of aberration at video rates, providing quasi-real time data processing [30].

As in many areas of science and engineering, deep learning strategies have been used to improve the potential of DHM technology. Deep learning and neural networks have enhanced the performance of traditional reconstruction algorithms. For example, deep learning has been utilized to automatically determine the in-focus reconstruction plane in DHM [31,32] and obtain color holographic microscope reconstructions [33]. Yin et al. proposed a deep learning framework for a reflection digital holographic setup without a microscopic imaging system (e.g., lensless configuration) in which the sample information was loaded on a phase-only spatial light modulator (e.g., non-biological samples) [34]. In 2019, Rivenson et al. presented a neural network model to retrieve the amplitude and phase reconstruction for lens-free DHM systems [35]. The same year, Wang et al. proposed a different approach for retrieving the complex information (e.g., amplitude and phase) of holograms using a neural network based on the Y-Net model [36] in common-path microscopy. In 2020, Vijayanagaram implemented a neural network model based on a U-Net architecture for numerical reconstruction of artificially generated in-line holograms [37]. Recently, Di et al. proposed a powerful strategy for quantitative phase image by implementing the PhaseNet convolutional neural network for reconstructing phase maps of size 128 × 128 recorded from a common-path digital holographic microscope in times of ~0.014 s (~71.43 fps) [38]. In [39], Moon et al. experimentally validated a conditional generative adversarial network (cGAN) for removing superimposed noise in the Gabor (in-line) holograms of red blood cells and elliptical cancer cells; the proposed cGAN model required the processing of off-axis DHM holograms before training the learning-based model. The aim of that work was not quantitative phase imaging in off-axis DHM. Ma et al. proposed a two-stage generative adversarial network (GAN) to provide accurate phase maps in DHM [40]. The two-stage GAN required a preprocessing stage to remove the interference fringes of the sample that were in the DHM hologram by background segmentation. Then, the vacancy sample area was inpainted with fringes generated by a deep learning algorithm prior to recovering the compensated phase maps. Since the implementation of the two-stage GAN was focused on a reflection (e.g., Michelson-based) DHM system, the model was not validated with biological samples. Although many studies have been reported involving deep learning models for holographic reconstruction, to the best of our knowledge, no learning-based method has been reported in the literature for full phase compensation in transmission off-axis DHM using biological samples. In this study, we propose a conditional generative adversarial network (cGAN) to reconstruct quantitative phase images free of aberration without the need for any pre- or post-numerical procedures, reducing the computational complexity of current reconstruction methods. The proposed learning-based model generated phase maps with stable backgrounds, immune to phase instabilities introduced by different experimental conditions during the recording stage. This paper is organized as follows: in Section 2, we describe the experimental setup, the creation of the dataset, and the structure of the cGAN model; in Section 3, the experimental results of the proposed model for static and dynamic red blood cells are discussed and we compare the performance of the model to a traditional automatic method and the U-Net model; finally, in Section 4, we summarize the main achievements of our research.

## 2. Materials and Methods

### 2.1. Experimental System

In this study, we used a traditional Mach–Zehnder off-axis DHM setup operating in transmission mode (Figure 1). The collimated beam emerging from a 532 nm He-Ne laser illuminates a beamsplitter that generates the object and reference wavefronts with the same power. The object wavefront illuminates the sample after being reflected by a plane mirror. Then, the light scattered by the specimen is collected by an imaging system comprising a 40X/0.65NA infinity-corrected microscope objective (MO) lens and a tube lens (TL). The MO and TL operate in the telecentric regime (e.g., the back focal plane of the MO lens coincides with the front focal plane of the TL) to optically compensate for the spherical wavefront introduced by the MO lens [41,42]. A magnified image of the sample is located at the back focal plane of the TL. The reference wave (e.g., a plane wave) propagates with no perturbations to a second beamsplitter that recombines both the object and reference wavefronts to generate the DHM hologram (i.e., the interference pattern produced by the superposition of both wavefronts). A CMOS camera, comprising 1920 × 1200 square pixels of 5.86 μm side, records this interference pattern. The sensor is located at the back focal plane of the TL so that image-plane conditions are met (i.e., an in-focus image of the sample is formed onto the sensor plane). In this DHM system, the off-axis configuration is achieved by changing the angle of the plane reference wavefront via tilting the mirror in the reference arm and the second beam splitter. The off-axis configuration ensures that amplitude and phase images can be reconstructed from a single hologram (i.e., single-shot DHM), making this DHM system more suitable for dynamic imaging.

The DHM hologram distribution, h(x,y), is expressed as
(1)$$h(x,y)={\left|r(x,y)\right|}^{2}+{\left|o(x,y)\right|}^{2}+o(x,y)r^{*}(x,y)+r(x,y)o^{*}(x,y),$$
where o(x,y) is the complex amplitude distribution of the object wavefront imaged by the telecentric imaging system, and r(x,y) is the complex amplitude distribution of the tilted plane wave. In Equation (1), the complex sample information, o(x,y), is encoded in the third and fourth terms of the hologram. The term o(x,y) should be isolated from the recorded hologram (Equation (1)) to reconstruct both the amplitude and phase images. Because the DHM system operates in an off-axis configuration, the first step in the computational processing involves the spatial filtering of the object spectrum in the spectrum of the recorded hologram (i.e., Fourier transform of the hologram), using a circular mask to select the object spatial frequencies [43]. The second step involves the phase compensation of the interfering angle introduced by the tilted reference wavefront. This compensation can be performed in the real space by multiplying the inverse Fourier transform of the filtered hologram spectrum, i.e., $o(x,y)\cdot r^{*}(x,y)$, and a digital replica of the reference wave, $r_{D}(x,y)$, commonly called the digital reference wavefront. In 2016, Trujillo et al. proposed a method based on a mean-thresholding-and-intensity-summation metric to generate the most trustable digital reference wave and reconstruct phase images with minimum phase aberrations [26]. Trujillo’s method was based on the observation that the best-reconstructed phase image (i.e., the reconstructed phase image without distortions) generated a white binary phase image; all pixels in the binary phase image using a fixed thresholding value should be one. Therefore, if a residual plane tilt distorts the reconstructed phase images, its binary image combines black and white pixels. The input parameters of the reconstruction method used were the image-plane holograms recorded in an off-axis telecentric DHM system, the source’s wavelength, and the sensor’s pixel size. The advantage of this method was the reconstruction of phase images with minimum phase perturbations. Although this and other computational methods are automatic and adaptable to different sample and imaging conditions, their performance is often hampered by artifacts such as background fluctuations and phase discontinuities unrelated to the specimens.

### 2.2. The Dataset

A microscopic slide containing normal (healthy) human red blood cells (RBCs) from Carolina Biological Supply Company (item # C25222, https://www.carolina.com/help/service-and-support?utm_source=homepage&utm_medium=banner&utm_campaign=whyshop_service (accessed on 20 November 2021)) was used to create our dataset to train and validate the proposed learning-based method. Although these RBCs present some absorption, they can be considered phase objects when no staining is applied. Unstained RBCs have been widely imaged in DHM, enabling the detection of malaria [44,45] and screening of diabetes [46] and sickle cell anemia [18,47] as well as other inherited anemias [48]. Our dataset was composed of raw DHM holograms and their corresponding reconstructed phase images [26]. The procedure to create this dataset is the following: We have recorded 30 different in-focus DHM holograms of the human RBCs in an off-axis Mach–Zehnder DHM system (Figure 1). The main proposal behind this work is to reconstruct DHM holograms without any computational processing, extracting the quantitative phase distribution directly from the raw in-focus hologram regardless of the orientation of the interference angle of the off-axis DHM system. Considering that a change in the interfering angle yields a different set of reconstruction parameters, the recorded 30 holograms were augmented via image orientation by rotating the holograms randomly in 90 or 180 degrees (Figure 2a). After the data augmentation was applied, the number of holograms increased to 65 RBCs holograms. We reconstructed the quantitative phase images of these holograms (Figure 2b), using the proposed automatic method by Trujillo et al. [26]. After reconstructing the holograms, each pair of holograms and reconstructed phase images was cropped into patches of 256 × 256 pixels. According to the camera specifications, each pair of holograms and reconstructed phase images was divided into 28 pairs of images (i.e., holograms and reconstructed phase images) to generate an experimental dataset of 1820 instances: 1512 pairs for the training dataset and 308 pairs for the validation dataset. A second validation dataset consisting of 7005 pairs of images was recorded to avoid similarities between the training and validation datasets due to the identical experimental conditions of the one-time acquisition process. This dataset was used to assess the effectiveness of the proposed model for different experimental conditions and to show that no overfitting occurs during the training of our proposed cGAN model. Finally, the cGAN model was tested to reconstruct quantitative phase images of dynamic DHM holograms. For this study, we mimicked the flow of red blood cells in a capillary by mounting a static microscopic sample of RBCs in a motorized translational stage. The velocity of the motor was such that the flow of the RBCs was 2.75 μm/s. We recorded a sequence of 300 holograms that tracked the flow of RBCs within blood plasma. It is important to highlight that all the recorded cells were in focus (e.g., the image plane condition was met for all cells). This experiment was completely equivalent to the experiment in which the capillary depth was smaller than the depth of field of the objective lens, which was approximately 0.69 μm, based on the manufacturer’s specifications.

### 2.3. The Proposed Learning-Based Method

We have proposed a learning-based strategy to avoid conventional computational processing while enabling video-rate quantitative phase imaging free of aberration. Our experimental RBC dataset was used to train a conditional generative adversarial network for image-to-image translation (pix2pix cGAN) [49]. The structure of this model is depicted in Figure 3. This learning-based model was comprised of two submodels: a generative model and a discriminator model. The generative model produces free-of-aberration phase maps, given DHM holograms as inputs. The supervised training of this submodel is defined by a weighted sum of two loss functions. The first function is an L1 loss which penalizes the difference between the generated reconstructed phase maps by the model G(x), and the target reconstructed phase maps by the traditional method y. The second function is the adversarial loss from the discriminator model. The discriminator model is trained to determine whether the generator has built an input phase map (i.e., yielding to a fake/artificial signal) or a target phase map (i.e., an actual signal). The factor λ is fixed as 100 to favor the L1 loss |G(x)−y| during training [49]. This parameter encourages the generator to produce plausible translations of the DHM holograms, not just plausible phase maps in the target domain. The models were trained with the Adam version of the stochastic gradient descent [50] with a small learning rate of 0.002 during 100 epochs.

In this cGAN architecture, the discriminator model is a deep convolutional neural network for binary classification. Each input to this model is a pair of an off-axis in-focus DHM hologram and its corresponding phase image concatenated horizontally in a 256 × 256 × 6 entries array. This patchGAN discriminator [51] processes 70 × 70 pixel regions of the input images, and the results of all patches are averaged to obtain an overall classification outcome for the input image. The model consists of six bidimensional convolutional layers in which the stride is fixed at 2 in each dimension to downsample the feature maps size after each convolution. The activation function of the first five layers is the LeakyReLu [52] to avoid blockage during training due to negative values in the input signals. In layers 2 to 5, batch normalization is also applied over every instance. The number of feature maps increases in each of these convolutional layers up to 512 to adjust the complexity of the model according to the data. In layer 6, the number of filters is finally reduced to synthesize the information to be learned by the model. This network was trained following two objectives: (1) to detect generated phase maps from DHM holograms by the generative model and (2) to detect target phase maps directly supplied to the discriminator. These two objectives were achieved by the model using two L1 loss functions mathematically represented by
(2)$$\begin{array}{l} L_{real}(w)=-y_{i\_real}\log(a_{i}(w))+(1-y_{i\_real})\log(1-a_{i}(w))=\log(L_{1}(real))\\ L_{fake}(w)=-y_{i\_fake}\log(a_{i}(w))+(1-y_{i\_fake})\log(1-a_{i}(w))=\log(L_{1}(fake)) \end{array}.$$

For each training step, these loss functions allow the tunning of the coefficients of the model (w) cooperatively using batches of target phase maps ($L_{real}$), and batches of generated phase maps ($L_{fake}$). The output of the patchGAN discriminator $a_{i}(w)$ is a 16 × 16 binary classification matrix of those when a generated or target phase map is detected, and zeros otherwise. In Equation (2), $y_{real}$ and $y_{fake}$ account for the labels of the data feed to the discriminator during training. Although two loss functions were used to train the discriminator, only the L1 loss over the real images was transferred in every training step to the generator model to update its coefficients. This loss value is known as the adversarial loss of the pix2pix cGAN model [49].

A U-Net architecture [53] was used as the generative model because it accurately reconstructs high spatial frequencies features [54]. The hallmark of the U-Net model is an accurate reconstruction of the phase values in those regions where the sample presents sharp jumps, such as the edges of the RBCs. The U-Net structure is based on a traditional bidimensional convolutional autoencoder for image-to-image translation with skip connections. The additional skip connections between asymmetrical layers in the U-Net structure guarantees that the bottleneck of the autoencoder structure can be circumvented. In other words, the skip connections shuttle the low-level information of the input DHM holograms directly across the network [54]. The model’s encoder part comprises eight convolutional layers with a 2-pixel stride in each dimension to downsample their feature size from 256 × 256 to 1 × 1. The number of filters in these convolutional layers was increased from 64 to 512 to account for the information from the datasets. The decoder part of the model is almost a mirrored version of the encoder. The differences between the encoder and decoder parts are the transpose bidimensional convolutional layers and the implemented dropout layers after its first four layers. The output in the generator model is a 256 × 256 × 3 entries array produced with a final hyperbolic tangent activation function which presents better results for different GAN architectures [55].

Since the composite loss functions of the GAN models do not converge, due to the adversarial fitting of the two involved sub-models [55], customized metrics must be used to guide the proper training of this learning-based proposal. Two metrics were selected in every epoch of the training stage to quantify the method’s performance at reconstructing phase maps without phase perturbations from raw DHM holograms. The first metric was a thresholding-and-summation metric (TSM) which accounted for phase discontinuities in every generated phase map. Whereas distorted phase images in DHM generate thresholded phase images with a mix of black and white pixels [26], the reconstructed phase image without phase aberrations (or minima phase perturbations in the presence of high-order aberrations such as astigmatism and coma) is the one whose thresholded phase image is white (i.e., all pixels in the thresholded phase image should be one) [26]. Therefore, our customized metric (TSM) counted the number of black pixels in the binary phase image (i.e., thresholded phase value equal to zero),
(3)$$\mathrm{TSM}[{\hat{h}}_{nor}(n,m)]=1-\frac{1}{N\cdot M}\sum_{n=0}^{N}\sum_{m=0}^{M}\mathrm{Thres}[{\hat{h}}_{nor}(n,m),0.2].$$

In Equation (3), ${\hat{h}}_{nor}$ stands for the normalized reconstructed phase map generated by the method in every training step; n and m are integer numbers running from 0 to N and M, respectively; N, and M are the number of pixels in each dimension of the output phase image, which coincides with the dimensions of the input hologram; Thres[·] represents a thresholding operator that converts each image pixel value larger than 0.2 into one, otherwise zero [26]. Figure 4a shows the plot of the average TSM values measured from reconstructed phase maps generated by the model for the full training (orange curve) and validation (blue curve) DHM holograms. Figure 4a shows that the normalized TSM for the validation dataset does not change considerably after the 10th epoch. Therefore, the network has converged in terms of the number of reconstructed phase discontinuities after the 10th epoch. Figure 4c shows a set of generated phase maps at different epochs for the same hologram of the validation dataset. The proposed model reconstructed the phase discontinuities correctly after the convergence has been reached.

The second customized metric involved measurement of the standard deviation (STD) inside one background region of the phase maps (i.e., an area in which the phase measurement is constant). For simplicity, we selected a region where no RBCs were present in the reconstructed phase images. An example of the region chosen to calculate the STD value is shown in Figure 4c; see the area enclosed by the blue square. This metric was calculated only for 15 generated phase maps per dataset, which provided a good representation of the STD metric for the whole dataset. Figure 4b plots the average STD values for the selected 15 phase maps per dataset. The STD metric allows quantifying the noise of the generated phase images, which must converge (i.e., the STD value is as low as possible) after the proposed method is adjusted correctly. According to this metric, the generated phase maps for the validation dataset present the minor noise level at the 12th epoch. After this point, the STD value starts increasing again. For the training dataset, the average STD metric also converges at the 12th epoch. Nonetheless, after this epoch, the model appears to start overfitting. Overfitting in supervised learning occurs when the tendency of the model’s performance on the training data diverges from the performance of the model on the validation data. The latter implies that the model is being fine-tuned to predict the training samples while at the same time it is failing to adequately predict the validation samples, which are the actual targets of the learning procedure. Therefore, considering that the model presented the least noise level at the 12th epoch, and the normalized TSM had already converged, we concluded that the best-fitted model coefficients for our learning-based method were found at this epoch (i.e., the 12th epoch).

## 3. Experimental Results

Once the generator was adjusted correctly within the GAN training, we used this learning-based model to reconstruct free-of-aberration phase images of static and dynamic human RBCs from experimental holograms recorded in an off-axis telecentric DHM system. To use our proposed learning-based method, a sample code to reconstruct quantitative free-of-aberration phase images from RBC holograms recorded in an off-axis telecentric-based DHM is freely available in [56]. The model in this repository contained the weights yielded during the training stage described in Section 2.3 and used in this section to retrieve the reported results. Therefore, any user can use this implementation to process off-axis diffraction-limited DHM recordings with no need for further fitting stages or robust pre- or post-processing procedures. Note that it is only required to match the field of view between the input hologram and the one used to train the network.

### 3.1. Experimental Quantitative Phase Images Obtained by the Proposed Learning-Based Model Using Static DHM Holograms

In our first experiment, we compared the performance of our learning-based cGAN model, after proper training, to that of the traditional reconstruction method [26] to analyze human RBCs in static conditions. Figure 5a–d shows four different holograms of the validation dataset. These holograms (Figure 5a–d) present different experimental conditions, such as the presence of dust in the sample, some defocusing effects, and changes in the background intensity. The latter may be related to temporal fluctuations in the experimental conditions from the illumination source and/or the implementation of the system. Whereas the panels in Figure 5e–h (second row) show the reconstructed phase images for the conventional method, panels in Figure 5i–l are the reconstructed phases for the proposed learning-based method (third row). According to these results (Figure 5), the performance of our proposed cGAN model is similar to the performance of the traditional reference method for the highest-quality reconstructed phase images performed by the conventional method (Figure 5e,h). However, the conventional method introduces some random phase distortions in some holograms, reconstructing low-quality phase images (Figure 5f,g). Contrarily, the cGAN model reconstructs these quantitative phase maps successfully without undesired phase distortions (Figure 5j,k). The colored arrows in these panels show other differences between the reconstructed phase images. For example, the green arrows indicate background regions in the reconstructed phase images with different phase values. Whereas the conventional method fails to quantify the background phase level correctly, the proposed method successfully provides a homogeneous background level for most testing DHM holograms. The blue arrows highlight RBCs whose phase distribution presents some distortions in the traditional method. The poor reconstruction performance of the conventional approach is related to the reliance on the estimated parameters of the digital reference wavefront to the temporal fluctuations in the experimental recording conditions. Although the performance of the cGAN method seems superior, the cGAN model introduces some blurring effects when reconstructing the contour of some RBCs, as the red arrows show.

For a better comparison of the differences between the reconstructed phase maps produced by the traditional reference method (Figure 5e–h) and the cGAN model (Figure 5i–l), we measured the two customized metrics described in Section 3, i.e., the TSM and STD. The values of these two metrics are reported in Figure 5. According to the reported TSM values, the cGAN model reconstructs phase images with lower TSM values than those obtained by the conventional method. Equation (3) shows that the best reconstructed phase image (i.e., phase image with minimum phase aberration) provides the lowest TSM value. Regarding the STD value, we measured this metric in the region highlighted by the yellow rectangle in panels Figure 5e–h. According to the reported STD values, the cGAN model reconstructs phase images with lower STD values than those obtained by the conventional method. These results allow us to conclude that the performance of the proposed learning-based method is better in terms of the quality of the reconstruction (i.e., minimum phase aberration and reduced background noise).

To further validate our method performance for these static holograms, we have compared the highest- and the lowest-quality reconstructed phase images provided by the conventional reference method to those obtained by our cGAN model. Figure 6 shows the unwrapped reconstructed phase maps and their corresponding 3D topographical view. We have used the Goldstein algorithm to unwrap the reconstructed phase images [57]. The unwrapped phase values (ϕ) have been converted in RBC’s thickness (*t*) via $\phi =2\pi \;t/[\lambda_{0}(n_{s}-n_{m})]$ where λ_0_ = 532 nm is the illumination wavelength, $n_{s}$ = 1.406 is the RBC refractive index [58], and $n_{m}$ = 1 is the surroundings’ refractive index. As Figure 6 shows, both methods provide similar results for the highest-quality reconstructed phase image reconstructed by the reference method (Figure 6a–c). However, the cGAN model achieves improved results for the lowest-quality phase map reconstructed by the conventional method, comparing panels (g)–(i) versus panels (j)–(l) in Figure 6. Note that the presence of phase aberrations in Figure 6g leads to reconstructing an RBC with negative optical thickness with respect to the background phase level. Quantitatively, the optical thickness of an RBC is around 0.8 µm, as was previously reported via a phase-shifting DHM [59]. Whereas the cGAN method provides RBCs with optical thickness around 0.8 μm, as shown by Figure 6f,l, the optical thickness of the RBCs in Figure 6i is around 1.5 µm. Apparent morphological changes are introduced by the erroneous reconstruction of the phase image provided by the reference method. This inaccurate result could lead to misleading sample identification, illness screening, or other diagnostics based on quantitative phase imaging.

### 3.2. Comparison of the Proposed Cgan Model against the U-Net Model and Validation of the Proposal’s Generalization Ability to System’s Diversity

A straightforward comparison of the performance of our method against that of the U-Net model is presented in this subsection to validate its appropriateness in the task of retrieving complete phase compensated phase maps in transmission off-axis DHM. The implemented U-net model has the same parameters as the generator model in our proposal, depicted inside the blue rectangle in Figure 3. The details of the generator model are described in Section 2.3. Overall, the training hyperparameters of this U-Net model are the same employed in our cGAN model generator. The main difference is the use of a logarithmic loss function to adjust the U-Net model weights. The U-Net model converged after 50 epochs. For each image in the validation dataset, this convergence was estimated by measuring the root-mean-square error (RMSE) between the reconstructed phase images of the conventional method and the one of the U-Net model at every epoch. Considering the same training dataset as the one for the cGAN model (1508 pairs of images), the measured RMSE on the validation data was 0.25. Since the input reconstructed phase images were normalized before training and prediction, this RMSE value means that there is an error of 25% between the target phase images and the reconstructed U-Net images. To minimize the RMSE error, we increased the training data set to 24,491 pairs of images. In this case, the convergence of the U-Net model was reached with an RMSE value of 0.04 at epoch 28. Figure 7 compares the reconstructed phase images obtained by the traditional method, the U-Net model (trained with 24,491 pairs of images), and our proposal (trained with 1512 pairs of images). Note that Figure 7 also shows the performance of the U-Net and proposed cGAN models using the new validation dataset to assess the goodness of these models for different experimental conditions and show that no overfitting occurs. The proposed cGAN model outperforms the U-Net model based on the reported TSM values even though the training dataset of the cGAN model is 16.2× smaller than the one used in the U-Net model. Note that the cGAN model reconstructs phase images with lower TSM values than those obtained by the U-Net model. In addition, the cGAN model retrieves the edges of the RBCs with greater accuracy than the U-Net model, as pointed out by the green circles and the orange arrows. To quantify the difference in the noise level, we have measured the STD values in the regions encircled by the dark blue lines in Figure 7. Based on these STD values, the U-Net model reconstructs phase images less sensitive to noise than the reconstructed cGAN phase images (e.g., lower STD values). Nonetheless, the STD values are approximately 2.24× lower than those provided by the conventional method. Although the cGAN model may introduce some remaining nuisances in the reconstructed phase images, as indicated by the red circles in Figure 7, overall, our method achieves better results than the traditional method and the U-Net model.

To validate the generalization ability of our proposed method in regard to system diversity, DHM holograms were recorded in a common-path DHM system using a Fresnel biprism. This DHM system was comprised of a 40X/0.74 Nikon MO lens, and the camera used had 5471 × 3648 square pixels of 2.4 μm side. The Fresnel-biprism-based DHM system still operated in the telecentric regime, and the camera was located at the microscope’s image plane. More details on this DHM system can be found in [60]. Note that a minimum preprocessing of the DHM holograms recorded by the Fresnel-based DHM system is required. The preprocessing step consisted of resizing the raw holograms to reduce their size by 40% and matching the field of view to those provided in the Mach–Zehnder-based holograms. After this step, we evaluated the traditional method and the U-Net and cGAN models to reconstruct these Fresnel-based DHM holograms. Figure 8 shows selected holograms from the Fresnel-based DHM system and their corresponding reconstructed phase images to test the generalization ability of the proposed cGAN model. As shown in Figure 8, the reconstructed phase images obtained by the traditional method present RBCs with a different phase value even though all cells are expected to yield the same phase values within experimental error margins. This phase difference in the RBCs is due to a phase difference of π introduced by the Fresnel biprism between each interfering wave. Nonetheless, both learning-based methods reconstruct all cells with the same phase values. The green arrow and the region enclosed by the red circle in Figure 8g,k illustrates that the U-Net model provides distorted details of RBCs. In contrast, the cGAN model reconstructs the specimen’s details accurately, as shown in the region inside the green circles and pointed out by the orange arrows. The measured STD values, computed for the region encircled by the dark blue circle, are similar for the three methods. In summary, the cGAN model can reconstruct quantitative phase images from off-axis DHM holograms with minima distortions, highlighted by the red circles in Figure 8, regardless of the optical configuration of the transmission DHM system.

### 3.3. Validation of the Proposed Learning-Based Model Using a Sequence of Dynamic DHM Holograms for Video-Rate Quantitative Phase Images

In this experiment, we investigate the performance and the robustness of the cGAN model for reconstructing aberration-free phase images in time-lapse DHM imaging. For this study, we recorded a sequence of 300 holograms that tracked the flow of RBCs within blood plasma. These recorded holograms were used as input images of our trained cGAN model. Figure 9 summarizes the performance of the traditional reference method [26] and the proposed learning-based model. In this figure, we have displayed five randomly selected phase images. The *n* value stands for the *i-*th frame in the time-lapse recording sequence. Whereas the RBC flow is marked by the blue arrow in Figure 9b, the yellow star is a visual aid to track the identical RBC across the time-lapse sequence. It can be seen that the traditional reference method reconstructs phase images with phase aberrations and varying background levels across the time-lapse sequences, see the red arrows in Figure 9a. In addition, the poor performance of the traditional method inhibits the proper visualization of inner structures in the dynamic imaged sample. In contrast, the performance of the cGAN model is superior, providing reconstructed RBCs images with fewer phase aberrations, homogeneous background (marked by the green arrows in Figure 9b), and reduced noise. However, our cGAN model does not exploit one of the major advantages of DHM, which is the reconstruction of quantitative phase images of defocused samples (e.g., cells, bacteria, and organisms located at different depths). In future work, we will investigate the performance of the cGAN model to reconstruct defocused holograms at different depths for several biological samples by implementing a microscopic slide with a flow path, enabling the recording of out-of-focus holograms.

Appendix A displays the rendering of the time-lapse reconstructed phase images for both the conventional method and the cGAN model. This video demonstrates the superior performance of our proposed method. Note that all the reconstructed phase images generated by the cGAN model present a homogenous background, resulting from the accurate compensation of the digital reference wavefront. In addition, the majority of inner structures in the imaged RBCs are retrieved without any phase distortion. Contrarily, the conventional method fails to reconstruct several frames in the time-lapse sequence, resulting in an unstable reconstruction technique. In summary, we estimated the number of frames per second (FPS) required for each method to reconstruct these phase images from the raw DHM holograms. For simplicity, we excluded the acquisition time in the estimation of the FPS. The processing time for each DHM reconstruction method is reported based on a laptop powered by an Intel Core I7 6700HQ CPU running at 2.60 GHz with 8 GB of RAM and hosting an NVIDIA Geforce GTX 960M GPU with 2 GB of RAM running at 1 GHz. Whereas the conventional method requires approximately 115 ms to reconstruct each raw hologram, the proposed fitted generator reconstructs each quantitative phase image in 14.8 ms on average, being 7.7× faster. According to these processing times, the video reconstruction is equal to 9 and 67 FPS for the conventional and the proposed method, respectively. The values of the FPS are also reported in Figure 7 and Appendix A.

## 4. Conclusions

In this study, we report on a conditional generative adversarial network (cGAN) to fully reconstruct quantitative phase images from human red blood cells (RBCs). To the best of our knowledge, this is the first learning-based method to reconstruct off-axis DHM holograms of biological samples with minimum phase distortions from raw holograms without the need for robust pre- or post-numerical procedures. The raw RBCs holograms were recorded using an off-axis DHM system operating at the telecentric regime. The proposed cGAN model was trained using two customized metrics specifically designed for tracking the imaging characteristics in DHM: (1) the number of phase discontinuities using a thresholding-and-summation metric (TSM, Equation (3)) and (2) the noise level measured in homogenous regions of the reconstructed phase maps using the standard deviation (STD). Because we used two customized metrics (i.e., TSM and STD), the proposed cGAN model converges rapidly (i.e., only 12 epochs are needed). This learning-based method was trained using 1512 pairs of raw holograms and their reconstructed phase images obtained by a traditional reference method [26]. Figure 5, Figure 6, Figure 7, Figure 8 and Figure 9 show the robustness of our cGAN method against the reference method for raw holograms with different experimental conditions. We also compared the performance of the cGAN model to the U-Net model in Figure 7 and Figure 8 to validate the superior performance of the cGAN model. In summary, the proposed cGAN method surpasses the setbacks of the reference method, resulting in quantitative phase images with reduced noise and constant background level. These improvements are consequences of the model used (i.e., conditional generative adversarial networks) since these supervised models were initially designed to the specific abstraction of low-level information. Additional advantages of the cGAN model are: (1) the retrieval of inner structures of the RBCs’ information and (2) its training time (approximately 2 h). The high performance of our cGAN model paves the way for video-rate quantitative phase imaging of dynamic studies using DHM. The main disadvantages of the proposed method are the data field of view and the cell density, which are reduced to 40 μm × 40 μm (256 × 256 pixels) and up to 9 cells for the field of view, respectively. We will increase the cell density and the field of view in future work by expanding the image size fed into the model. In addition, although the cGAN model was validated using RBCs, it could be straightforwardly extended to reconstruct quantitative phase images of any type of dynamic biological sample. Future studies will upgrade the training dataset, including anemic RBCs, glioblastoma cells, and diatoms. The limitation of the cGAN model is that this method only works for off-axis DHM systems operating in the telecentric regime. Future studies will address this limitation by upgrading the model to ensure accurate quantitative phase analysis applicable to any off-axis DHM system.

## Figures and Tables

**Figure 1 sensors-21-08021-f001:**
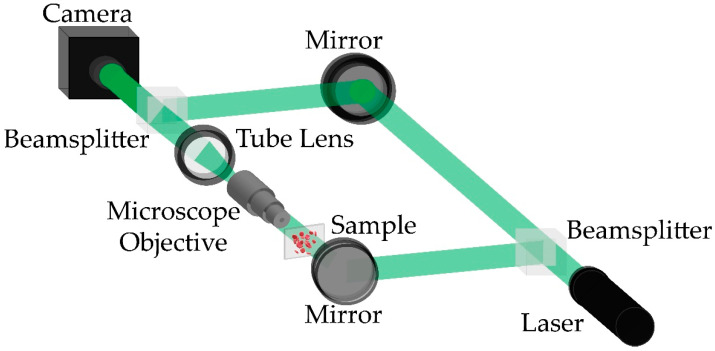
Conventional off-axis Mach–Zehnder DHM setup operating in telecentric regime.

**Figure 2 sensors-21-08021-f002:**
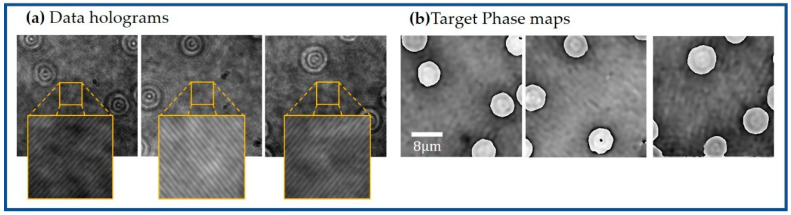
Illustration of the RBC dataset: (**a**) Input holograms after image orientation; (**b**) their respective phase maps reconstruction.

**Figure 3 sensors-21-08021-f003:**
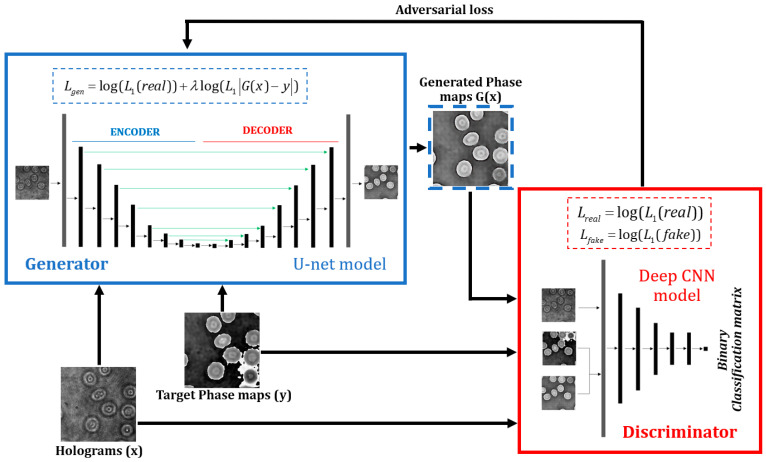
Structure of the image-to-image translation conditional generative adversarial network (pix2pix cGAN) for reconstructing quantitative phase images in off-axis DHM. See text for further details.

**Figure 4 sensors-21-08021-f004:**
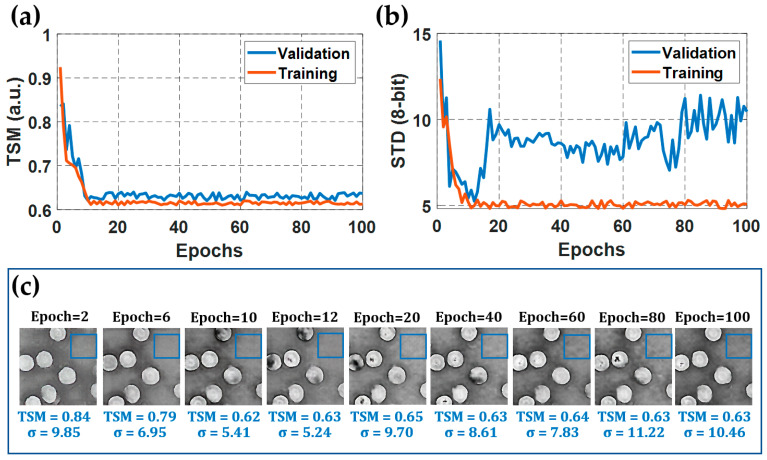
Performance of the cGAN model for the training and validation dataset: (**a**) The average TSM value of the reconstructed phase maps measured for the complete validation and training dataset; (**b**) the average STD values for the background regions of the selected 15 phase images per dataset; (**c**) reconstructed phase maps of the identical hologram of the validation dataset provided by the proposed cGAN model at different epochs.

**Figure 5 sensors-21-08021-f005:**
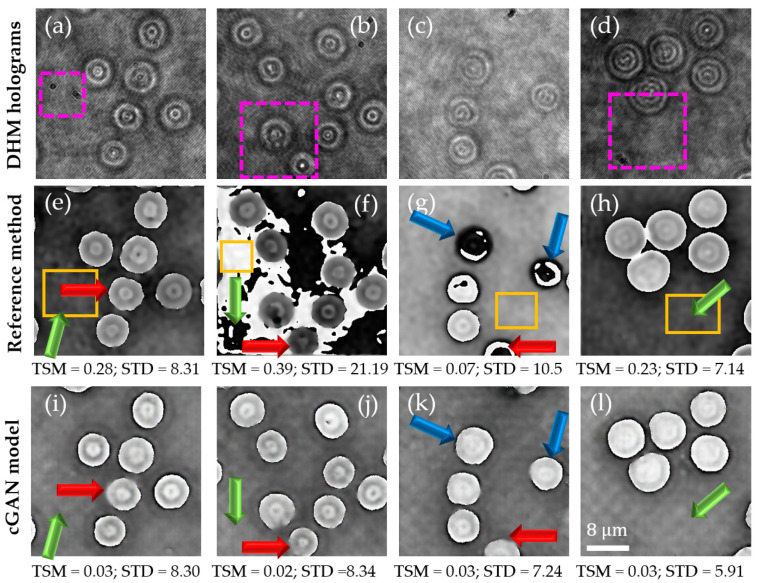
Results of the proposed learning-based method for aberration-free phase reconstruction of RBC samples in DHM. Panels (**a**–**d**) are the DHM holograms, illustrating different experimental conditions., marked by magenta dashed rectangles, such as dust in the sample, some defocusing effects, and changes in the background intensity. Panels (**e**–**h**) are the results obtained via the conventional method. Panels (**i**–**l**) show the phase map obtained by our proposal. All phase maps presented in this figure are accompanied by their STD and TSM values. The yellow rectangle in panels (**e**–**h**) mark the region in which we have estimated the STD value. The colored arrows in panels (**e**–**l**) show differences between the reconstructed phase images; see text for further details.

**Figure 6 sensors-21-08021-f006:**
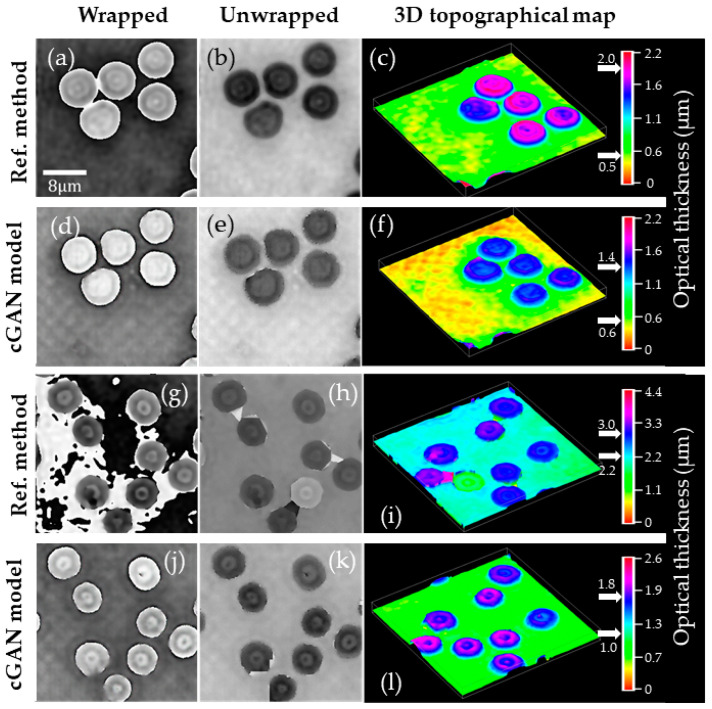
Quantitative comparison between the highest (**a**–**f**) and the lowest (**g**–**l**)-quality reconstructed phase images provided by the conventional reference method (first and third rows) to those obtained by our cGAN model (second and forth rows). The first and second column displays the wrapped and unwrapped reconstructed phase images, respectively. The third column shows the three-dimensional (3D) pseudocolor distribution of the optical thickness.

**Figure 7 sensors-21-08021-f007:**
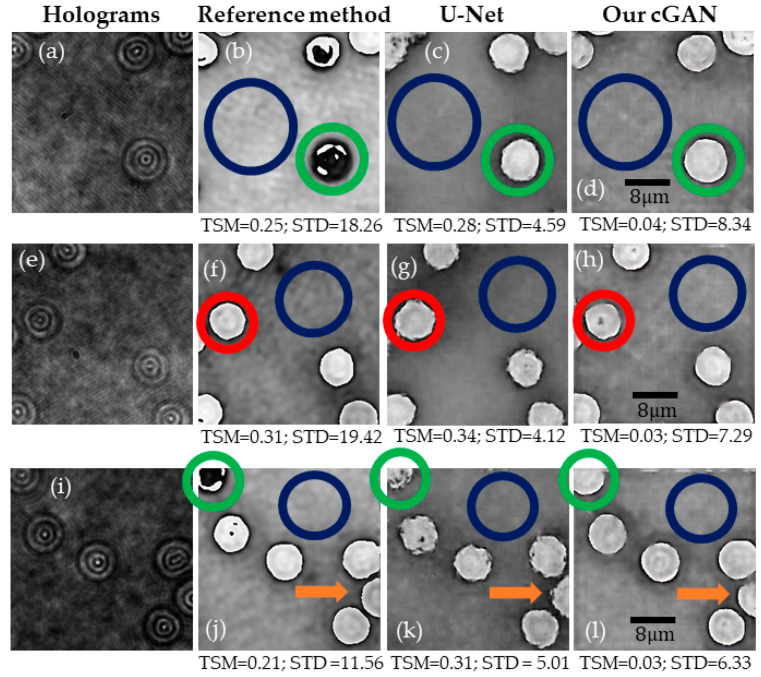
Comparison between the U-Net model and our cGAN model. The first column displays the selected DHM holograms from the validation dataset (panels (**a**,**e**,**i**)). The reconstructed phase images obtained by the conventional method are illustrated in the second column (panels (**b**,**f**,**j**)). Columns 3 (panels (**c**,**g**,**k**)) and 4 (panels (**d**,**h**,**l**)) show the reconstructed phase images achieved by the U-Net and our cGAN model. The colored circles and arrows show differences between the reconstructed phase images; see text for further details.

**Figure 8 sensors-21-08021-f008:**
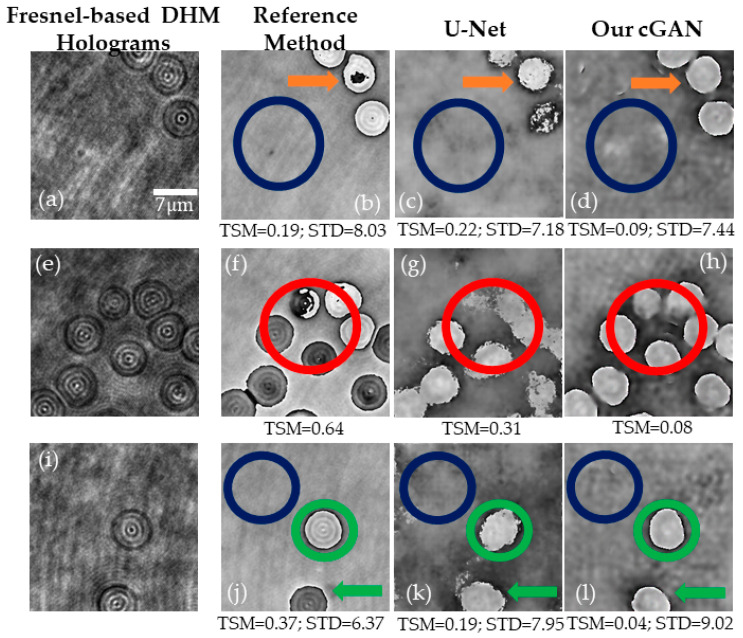
Evaluation of the generalization of the cGAN model to a common-path DHM system. The first column (panels (**a**,**e**,**i**)) shows selected DHM holograms recorded using a Fresnel-based DHM system. Columns 2, 3, and 4 display the reconstructed phase images obtained by the conventional method (panels (**b**,**f**,**j**)), the U-Net model (panels (**c**,**g**,**k**)), and the proposed cGAN model (panels (**d**,**h**,**l**)), respectively. The dark blue circle mark the region in which we have estimated the STD value. The colored circles and arrows show differences between the reconstructed phase images; see text for further details.

**Figure 9 sensors-21-08021-f009:**
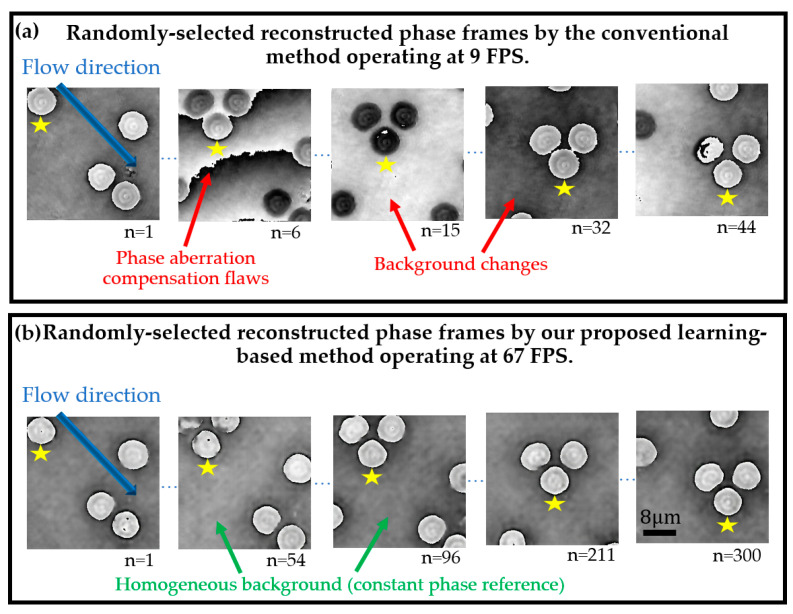
Evaluation of the traditional reference and GAN models to reconstruct aberration-free phase images in time-lapse DHM imaging. See Appendix A for video rendering of the reconstructed phase images of the flow of RBCs. The blue arrow marks the flow direction. The yellow star marks the identical RBC across the time-lapse sequence. The red and green arrows show differences between the reconstructed phase images.

## Data Availability

The weights yielded by our proposed cGAN model after the training stage are freely available in GitHub [56] to reconstruct quantitative phase images from RBC holograms recorded in an off-axis telecentric-based DHM.

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
