# Peer review of "Video-Rate Quantitative Phase Imaging Using a Digital Holographic Microscope and a Generative Adversarial Network"

_sensors, 2021, doi:10.3390/s21238021_

Round 1

Reviewer 1 Report

The authors propose a conditional generative adversarial network (cGAN) for robust and fast quantitative phase imaging in DHM. The reconstructed phase images provided by the GAN model present stable background levels, enhancing the visualization of the specimens for different experimental conditions in which the conventional approach often fails. After proper training, the proposed GAN yields to a computationally efficient method, reconstructing DHM images 7X faster than conventional computational approaches. The paper does not show itself to make a convincing case for novelty and impact. There are some Comments as follows existing. This manuscript could be considered for publication with major modifications at least.

Comments:

  1. Compared with the deep learning-based holography, the authors only use the cGAN, instead of U-net, to reconstruct the hologram. In fact, many works have deeply combined GAN with holographic reconstruction for improving [1-4]. The novelty of this manuscript is not enough.

[1] Yin D, Gu Z, Zhang Y, et al. Digital Holographic reconstruction based on deep learning framework with unpaired data[J]. IEEE Photonics Journal, 2019, 12(2): 1-12.

[2] Moon I, Jaferzadeh K, Kim Y, et al. Noise-free quantitative phase imaging in Gabor holography with conditional generative adversarial network[J]. Optics Express, 2020, 28(18): 26284-26301.

[3] Khan A, Zhijiang Z, Yu Y, et al. GAN-Holo: Generative Adversarial Networks-Based Generated Holography Using Deep Learning[J]. Complexity, 2021, 2021.

[4] Ma S, Liu Q, Yu Y, et al. Quantitative phase imaging in digital holographic microscopy based on image inpainting using a two-stage generative adversarial network[J]. Optics Express, 2021, 29(16): 24928-24946.

  1. The manuscript only compares the traditional reconstruction method with cGAN method. The Methods based on deep learning, such as U-net, should also be compared under the same conditions.
  2. The authors analyze the validation set data to show the effect of cGAN. However, the validation set and the training set are similar from one-time acquisition process. In training of cGAN, the performance on validation set also participates in human judgment to prevent the over-fitting. Therefore, the data is not convincing. The author should recollect data again to test the actual effect of cGAN.
  3. The authors claim that “any user can straightforwardly use this implementation to process DHM recordings with no need for further fitting stages or additional pre- or post-processing procedures provided the DHM system follows the same optical configuration as the one reported in Section 2.1”. It is ridiculous, because the manuscript does not provide the generalization ability for sample and system diversity. As long as the state of the system changes, the effect of cGAN will be greatly reduced. To support this claim, more evidence is needed in the article.
  4. The author emphasizes that the processing speed of cGAN is ×7 times faster than traditional methods. In fact, any deep learning-based methods can achieve this speed. The authors should pay more attention to the disadvantages of deep learning methods, such as the generalization ability, the data volume and the training time, and make improvements in this area.

Author Response

The reviewer's comments are addressed in the attached cover letter.

Reviewer 2 Report

This paper reports a new, computationally less intensive method for quantitative phase analysis from DHM. While potentially interesting, there is a major defect/oversight in the manuscript. The Methods report in great detail the sample used for the static holograms, but do not even mention that holograms "in blood plasma" were obtained. This comes as a surprise in Section 3.2, where results are discussed.

This method will not be particularly useful if it's only good for samples immobilized on a slide. Having motion and depth, and cells at different degrees of defocus, is the real point of DHM. So these blood plasma experiments are critical to the value of the paper. In particular:

--what was the sample chamber design, style, depth? Are the cells constrained to a capillary or free to float?

--flow rate?

--cell density?

This whole segment seemed "tacked on" at the end even though it is absolutely key to the value of the paper.

Author Response

We appreciate the reviewer’s time in reading our manuscript. While we agree that the dynamic results highlight the novelty and impact of our work, the focus of this manuscript is the investigation and evaluation of the cGAN algorithm as a real-time reconstruction method in DHM. In comparison with the previously reported learning-based methods, this is the first learning-based method to reconstruct quantitative phase images of biological samples with minimum phase distortions from raw holograms without the need for any robust pre- or post-numerical procedure. The raw RBCs holograms were recorded using an off-axis DHM system operating at the telecentric regime. Because the proposed cGAN model was trained using two customized metrics (e.g., TSM and STD) that track the imaging characteristics in DHM, the proposed cGAN model converges rapidly. In addition to comparing this learning-based model to the traditional reference method and the U-Net model in the revised manuscript, we have also validated its performance for reconstructing phase images recorded in a common-path DHM system. Please read response of the comments made by Reviewer 1. The main disadvantages of the proposed method are the data field of view and the cell density which are reduced to 40×40 μm2 and medium (up to 9 cells for the field of view), respectively. Future work will address these limitations and investigate the performance of the cGAN model to reconstruct quantitative phase images from defocused holograms.

The Methods’ section has been revised to include a description of dynamic DHM holograms for video-rate quantitative phase images. Regrettably, at present, our experimental laboratories do not have the required equipment to record the flow of red blood cells in a capillary or free to float. Instead, we mimicked this dynamic scenario by mounting a static microscopic sample of RBCs in a motorized translational stage. The velocity of the motor was such that the RBCs’ flow was 2.75 μm/s. It is important to highlight that all the recorded cells were in focus. This experiment would be completely equivalent if the capillary depth is smaller than the depth of field of the objective lens, which is approximately 0.69 μm based on the manufacturer’s specifications. However, we agree that one of the major advantages of DHM is the reconstruction of quantitative phase images of defocused samples (e.g., cells, bacteria, and organisms are different depths). In future work, we will investigate the performance of the cGAN model to reconstruct defocused holograms at different depth for several biological samples, including anemic and normal RBCs, glioblastoma cells, and diatoms. Also, in collaboration with our biological collaborators, we will implement a microscopic slide that will have a flow path, enabling the recording of out-of-focus holograms. 

The following text has been added in the subsection 2.2 of the revised manuscript:

“Finally, the cGAN model has been tested to reconstruct quantitative phase images of dynamic DHM holograms. For this study, we mimicked the flow of red blood cells in a capillary by mounting a static microscopic sample of RBCs in a motorized translational stage. The velocity of the motor was such that the RBCs’ flow was 2.75 μm/s. We have recorded a sequence of 300 holograms that tracks the flow of RBCs within blood plasma. It is important to highlight that all the recorded cells were in focus (e.g., the image plane condition was met for all cells). ). This experiment would be completely equivalent to the one in which the capillary depth is smaller than the depth of field of the objective lens, which is approximately 0.69 μm based on the manufacturer’s specifications.”

The following text has been added in the subsection 3.3 of the revised manuscript:

“However, our cGAN model does not exploit one of the major advantages of DHM, which is the reconstruction of quantitative phase images of defocused samples (e.g., cells, bacteria, and organisms located at different depths). In future work, we will investigate the performance of the cGAN model to reconstruct defocused holograms at different depths for several biological samples by implementing a microscopic slide with a flow path, enabling the recording of out-of-focus holograms.”

Round 2

Reviewer 1 Report

the experimental results can be improved.

Reviewer 2 Report

The authors have addressed my concerns, though it really isn't clear if this method is useful.